# Vitamin D_3_ Metabolic Enzymes in Plateau Zokor (*Myospalax baileyi*) and Plateau Pika (*Ochotona curzoniae*): Expression and Response to Hypoxia

**DOI:** 10.3390/ani12182371

**Published:** 2022-09-11

**Authors:** Xiaoqi Chen, Zhifang An, Linna Wei, Jiayu Zhang, Jimei Li, Zhijie Wang, Conghui Gao, Dengbang Wei

**Affiliations:** 1Research Center for High Altitude Medicine, Qinghai University, Xining 810016, China; 2State Key Laboratory of Plateau Ecology and Agriculture, Qinghai University, Xining 810016, China

**Keywords:** vitamin D_3_, lithocholic acid, hypoxia, plateau zokor (*Myospalax baileyi*), plateau pika (*Ochotona curzoniae*), Qinghai-Tibet Plateau

## Abstract

**Simple Summary:**

Vitamin D_3_ is mainly synthesized in the skin after exposure to sunlight and plays a fundamental role in the absorption of calcium and phosphate. Lithocholic acid increases intestinal calcium absorption in vitamin D-deficient rats. The plateau zokor, which lives underground throughout its life, and plateau pika are unique animal species on the Qinghai-Tibet Plateau. To investigate how animals that are rarely exposed to sunlight deal with the expected deficit in vitamin D_3_ levels and the effect of hypoxia on vitamin D_3_ metabolism, the levels of key biomarkers related to vitamin D_3_ metabolism, and the mRNA and protein levels of enzymes involved in key pathways were measured. Plateau zokor was deficient in vitamin D_3_ and has a higher level of lithocholic acid, but its calcium levels were within the normal range. The levels of vitamin D_3_ and the expression levels of vitamin D_3_ metabolism-related genes *CYP2R1* and *CYP27B1* decreased with increasing altitude in plateau pika. Plateau zokor was deficient in vitamin D_3_, and lithocholic acid appears to substitute it. In addition, hypoxia suppresses vitamin D_3_ in plateau pika by down-regulating the expression of vitamin D_3_ metabolism-related genes.

**Abstract:**

Vitamin D_3_ (D_3_) is produced endogenously from 7-dehydrocholesterol by irradiation and is an important secosteroid for the absorption of calcium and phosphate. Lithocholic acid (LCA) increases intestinal paracellular calcium absorption in a vitamin D receptor-dependent manner in vitamin D-deficient rats. The plateau zokor (*Myospalax baileyi*), a strictly subterranean species, and plateau pika are endemic to the Qinghai-Tibet Plateau. To verify whether the zokors were deficient in D_3_ and reveal the effects of hypoxia on D_3_ metabolism in the zokors and pikas, we measured the levels of 25(OH)D_3_, calcium, and LCA, and quantified the expression levels of D_3_ metabolism-related genes. The results showed an undetectable serum level of 25(OH)D_3_ and a significantly higher concentration of LCA in the serum of plateau zokor, but its calcium concentration was within the normal range compared with that of plateau pika and Sprague-Dawley rats. With increasing altitude, the serum 25(OH)D_3_ levels in plateau pika decreased significantly, and the mRNA and protein levels of CYP2R1 (in the liver) and CYP27B1 (in the kidney) in plateau pika decreased significantly. Our results indicate that plateau zokors were deficient in D_3_ and abundant in LCA, which might be a substitution of D_3_ in the zokor. Furthermore, hypoxia suppresses the metabolism of D_3_ by down-regulating the expression of *CYP2R1* and *CYP27B1* in plateau pika.

## 1. Introduction

Vitamin D_3_ is a fat-soluble secosteroid that has a fundamental role in the maintenance of calcium and phosphate homeostasis [1,2,3,4,5]. Vitamin D_3_ has a variety of pleiotropic functions in many extra-skeletal targets [2]. For instance, it regulates DNA stability, cell proliferation and differentiation, and innate and adaptive immunity [2,6,7,8]. Increased intestinal absorption of ingested calcium is vitamin D_3_′s key function in maintaining calcium homeostasis [4]. Regarding the absorption sites involved, the distal intestine, in addition to the duodenum, which only absorbs 8–10% calcium, plays an important role in intestinal calcium absorption [9,10,11,12]. Vitamin D_3_ promotes intestinal calcium absorption mainly via the transcellular pathway by a complex network of calcium-regulating substances, of which transient receptor potential vanilloid 6 (TRPV6) and calcium-binding protein D_9k_ (calbindin-D_9k_) have been identified as the major intestinal targets and paracellular pathways, including cadherin-17, claudin-12, claudin-2, and claudin-1; however, the role of vitamin D_3_ in the regulation of passive calcium transport remains a matter of debate [2,4,13,14,15,16,17,18,19]. Thus, the mechanisms by which vitamin D_3_ regulates intestinal calcium absorption have remain unclear.

About 80% of vitamin D_3_ is endogenously produced in the skin, and the remaining 20% comes from food [20]. However, the majority of foods contain only small amounts of vitamin D_3_, and cutaneous synthesis is the main source of vitamin D_3_. Vitamin D_3_ is produced in the skin from its substrate 7-dehydrocholesterol (7-DHC) [21], an intermediate in cholesterol synthesis. Following an electrocyclic rearrangement of the ring at the C9 and C10 positions, 7-dehydrocholesterol is transformed into pre-vitamin D_3_ under the influence of ultraviolet B (UVB) radiation, which then undergoes thermal isomerization to vitamin D_3_ after rearranging the triene structure of the molecule [22,23]. This is a non-enzymatic process, but must be produced by irradiation with UV light and heat; vitamin D_3_ is not biologically active by itself [2]. Vitamin D_3_ undergoes two subsequent hydroxylation reactions to produce the active form 1, 25(OH)_2_D_3_. First, vitamin D_3_ is metabolized to 25-hydroxyvitamin D_3_ [25(OH)D_3_] mainly by CYP2R1 in the liver [24,25]. The only 1α-hydroxylase, CYP27B1, then transforms 25(OH)D_3_ into 1, 25-dihydroxy vitamin D_3_ [1, 25(OH)_2_D_3_] in the kidneys [26]. The total serum concentration of 25(OH)D_3_ is thought to be the best indicator of vitamin D_3_ status [27,28]. CYP24A1 is a catabolic enzyme that can convert 25(OH)D_3_ and 1, 25(OH)_2_D_3_ to 24, 25(OH)_2_D_3_ and 1, 24, 25(OH)_3_D_3_, respectively, via a negative feedback mechanism in vitamin D_3_ signaling, and is a helpful marker of vitamin D_3_ catabolism [29]. Vitamin D deficiency is also associated with low serum calcium levels, leading to musculoskeletal effects [30]. Vitamin D deficiency leads to rickets in children and osteoporosis in adults, resulting in poor bone mineralization [5]. In nature, a variety of rodents such as the naked mole-rat, Damara mole-rat, and plateau zokor live entirely underground, which effectively shields them from sunlight, and feed on an herbivorous diet [31,32,33]. Previous studies have shown that the subterranean rodents Damara mole-rat and naked mole-rat have a natural vitamin D-deficient status [34,35]. Therefore, sunlight is essential for the photolysis of 7-dehydrocholesterol to vitamin D_3_. Recently, a study showed that the biosynthesis of steroid hormones is activated in response to hypoxia in Sprague-Dawley (SD) rats [36]. Vitamin D plays a role in preventing and protecting lung injury in hypoxia tolerance. A previous study demonstrated that hypoxic sheep had higher circulating concentrations of 25(OH)D and 1, 25(OH)_2_D than normoxic sheep [37]. However, one study reported an alarmingly low level of 25(OH)D among nomads in Tibet [38]. Therefore, the effect of hypoxia on vitamin D_3_ metabolism has not been elucidated.

1, 25(OH)_2_D_3_ is a physiological ligand of the vitamin D receptor (VDR). Bile acids are steroid acids produced by hepatocytes from cholesterol and circulate between the liver and the intestine [39]. Primary bile acids, cholic acid (CA), chenodeoxycholic acid (CDCA), and their respective secondary bile acids, deoxycholic acid (DCA) and lithocholic acid (LCA), are the four major bile acids. LCA reportedly activates VDR in vitro [40,41,42]. A study found that LCA not only increases serum calcium levels, but also stimulates 24-hydroxylase expression in the kidney and acts on the bone to mobilize calcium, only in vitamin D-deficient rats [41]. Therefore, LCA may be a substitute for vitamin D_3_ for calcium homeostasis in vitamin D-deficient animals.

The Qinghai-Tibet Plateau is the highest and largest plateau on Earth. Hypoxia, strong ultraviolet radiation, and low temperatures are the major challenges at high altitudes, with hypoxia being the most obvious characteristic of the climate [43,44,45]. The plateau zokor (*Myospalax baileyi*) is a unique subterranean rodent in the Qinghai-Tibet Plateau that lives underground throughout its life [46]. Plateau pika (*Ochotona curzoniae*), an indigenous species of the Qinghai-Tibet Plateau, is a tiny, non-hibernating mammal that inhabits meadows at altitudes of 3000–5000 m [47,48,49]. Over long-periods of adaption and evolution, the plateau zokor and plateau pika have developed a variety of strategies to physiologically and genomically adjust to the hypoxic environment and become highly advanced hypoxia-tolerant animals [50,51,52]. Based on the above analysis, we speculated that the plateau zokor might have a naturally impoverished vitamin D_3_ state owing to the lack of sunlight, and LCA might be a substitute for vitamin D_3_. Moreover, whether hypoxia upregulates or inhibits vitamin D_3_ metabolism is currently unclear. Therefore, the contents of 25(OH)D_3_, calcium, and lithocholic acid of plateau zokor, plateau pika, and SD rats were quantified, and their mRNA and protein levels of vitamin D_3_ metabolism-related genes in the liver and kidney were evaluated. These indicators were compared in plateau zokor and plateau pika at different altitudes.

## 2. Materials and Methods

### 2.1. Animal Procedures

High-altitude group plateau zokors and high-altitude group plateau pikas were captured at an altitude of 3700 m (36°72′ N, 101°28′ E) at Laji Mountain in Guide County, Qinghai Province, China. The average oxygen level in the sampling area was 180.6 g/m^3^. Low-altitude group plateau zokors were captured at an altitude of 2700 m (36°26′ N, 101°69′ E) at Laji Mountain in Guide County, Qinghai Province, China. The average oxygen level in the sampling area was 212.6 g/m^3^. Low-altitude plateau pikas were captured in the Qinghai Lake area (37°16′ N, 99°52′ E; 3200 m) in Gangcha County, Qinghai Province, China, where the average oxygen level was 196.6 g/m^3^. Zokor and pika were live-trapped in the summer months (June and July). To avoid differences due to stimulation, the animals were placed in their cages for 4–5 h of silence and the zokor was placed in dark. The Sprague-Dawley rats were purchased from the Experimental Animal Center of Lanzhou University (36°02′ N, 103°51′ E; 1500 m), Lanzhou City, Gansu Province, China. The average oxygen level in Lanzhou City was 251.0 g/m^3^. The SD rats from Lanzhou (1500 m) were kept and moved to Xining (2200 m) one week in advance for adaptation to the lab environment. In order to avoid variation, the rats were fed adaptively for 1 week, subjected to a 12–12 h circadian rhythm, provided suitable feed, water, and bedding, and their hair, excrement, and activity were observed. All the animals were male, and the weights of the zokors, pikas, and SD rats were 200–250 g, 100–150 g, and 200–250 g, respectively. Six animals from each group were used to quantify 25(OH)D_3_ and LCA, and nine were used to quantify calcium and determine the expression levels of the vitamin D metabolism-related-genes.

Sample collection was conducted at same time (14:00–16:00) for the zokor, pika, and SD rats to minimize circadian rhythm effects. After the animals were anesthetized in situ with chloral hydrate (10%), 8 mL of blood was collected from the carotid artery in foil-wrapped tubes without anticoagulants. Care was taken disinfect the animal during blood collection to prevent microbial contamination. The blood samples were left for 40 min, then centrifuged and the supernatant collected. The samples were then stored in liquid nitrogen. Abdominal aortic blood samples were collected for calcium concentration analysis. The liver and kidneys were placed in liquid nitrogen. All the procedures were performed according to the China Practice for the Care and Use of Laboratory Animals and approved by the China Zoological Society (permit number: GB/T35892-2018).

### 2.2. Ultra Performance Liquid Chromatography-Tandem Mass Spectrometry (UPLC-MS/MS)

Analytical standard for 25-hydroxy D_3_ was purchased from Sigma-Aldrich (St. Louis, MO, USA). 4-phenyl-1, 2, 4-triazoline-3, and 5-dione and analytical standards of LCA were provided by Shanghai Yuanye Technology Development Co., Ltd. (Shanghai, China). The purities of the standards were >98%. The LC-MS-grade solvents acetonitrile, formic acid, and methanol were purchased from Merck (Darmstadt, Germany). Deionized water was obtained using a Millipore-Q system (Millipore, Billerica, MA, USA).

#### 2.2.1. Quantitative Study of 25(OH)D_3_

Chromatographic separation was carried out on an Acquity UPLC (Waters, Milford, MA, USA) HSS T3 (100 mm × 2.1 mm, 1.8 μm) column operating in a programmed gradient mode at a flow rate of 0.3 mL min^−1^ using mobile phase A: formic acid (0.1%) in water and phase B: formic acid (0.1%) in methanol. The gradient elution program was optimized for the separation, and the program was as follows: 0–2 min, 50% B; 2–5 min, 100% B; and 5–6 min, 50% B. The temperature of the auto-sampler before analysis was 4 °C and the injection volume was 15 μL.

Mass spectrometry was performed on an AB SCIEX 5500 QQQ-MS (AB Sciex, CA, USA) equipped with an electrospray ionization (ESI) interface. The tandem mass spectrometer was operated in multiple reaction-monitoring modes. The optimal parameters were as follows: ion spray voltage, 4500 V; and turbo spray temperature, 450 °C. The flows of the curtain, collision, and ion source gases were set to 35, 9, and 55 psi, respectively. 

The thawed samples were vortexed for 10 s, methanol/dichloromethane (1:2, *v*/*v*) was added and sonicated at 4 °C for 30 min. The samples were incubated at 4 °C for 60 min, then centrifuged and dried in a concentrator. PTAD (4-phenyl-1, 2, 4-triazoline-3, 5-dione) solution (0.25 mg/mL) was added to the dried samples and reacted for 1 h in the dark. After derivatization was completed, the sample was placed in a concentrator to dry, reconstituted with 50% acetonitrile-water, and the supernatant was collected for UPLC-MS/MS analysis. The prepared standard solution was added to the injection bottle, and the retention time of the substance peak was determined. Integration was performed using MultiQuant software, and content calculation was performed using a standard curve.

#### 2.2.2. Quantitative Study of LCA

Chromatographic separation was carried out on an Acquity UPLC BEH C18 (100 × 2.1 mm, 1.7 μm) column. A programmed gradient mode of mobile phase A: formic acid (0.01%) in water and phase B: formic acid (0.01%) in acetonitrile was run at a flow rate of 0.3 mL min^−1^. The gradient elution program was optimized for the separation, and the program was as follows: 0–2 min, 10% B; 2–5 min, 40% B; 5–7.5 min, 45% B; 7.5–9.5 min, 60% B; 9.5–11.5 min, 80% B; 11.5–14 min, 80% B; and 14–16 min, 10% B. The temperature of the auto-sampler before analysis was 4 °C and the injection volume was 4 μL.

Mass spectrometry was as described for quantitative study of 25(OH)D_3_. The optimal parameters were as follows: ion spray voltage, −4200 V; and turbo spray temperature, 450 °C. The flows of the curtain, collision, and ion source gases were set to 35, 7, and 35 psi, respectively. 

Thawed samples were vortexed for 10 s, and methanol was added to the sample (6:1, *v*/*v*), and sonicated at 4 °C for 30 min. The sample was centrifuged and dried in a concentrator. Then, it was reconstituted with 200 μL of methanol, and the supernatant was collected for UPLC-MS/MS analysis. The prepared standard solution was added to the injection bottle, and the retention time of the substance peak was determined. Integration was performed using MultiQuant software and content calculation was performed using the internal standard method.

### 2.3. Calcium Concentration Analysis

The arterial calcium levels of the plateau zokor, plateau pika, and SD rats were measured using an i-STAT analyzer (Abbott Point of Care Inc., Princeton, NJ, USA), and the measurement temperature was set at 37 °C. The reference range for calcium concentration was 0.25–2.50 mmol/L.

### 2.4. Total RNA Isolation and qRT-PCR

Total RNA was extracted from the liver and kidney of *Myospalax baileyi* using the TRIzol reagent (Invitrogen, Carlsbad, CA, USA). The RNA concentration was measured using an ultraviolet spectrophotometer. Next, 4 μg of total RNA was reverse-transcribed to first-strand cDNA using FastKing gDNA Dispelling RT SuperMix (Tiangen, Beijing, China). The mRNA expression levels of *CYP2R1*, *CYP27B1*, and *CYP24A1* were evaluated by qRT-PCR, as described previously [53]. The nucleotide sequences of the primers used for the qPCR are listed in Table 1.

### 2.5. Western Blotting

Polyclonal antibodies against CYP2R1, CYP27B1, and CYP24A1 of plateau zokor and plateau pika were produced by Beijing Liuhe Huada Gene Technology Co., Ltd. (Beijing, China). SD rat antibodies against CYP2R1 (Invitrogen, PA5-87927), CYP27B1 (Invitrogen, PA5-79128), and CYP24A1 (Invitrogen, PA5-79127) were purchased from Invitrogen (Carlsbad, CA, USA). Polyclonal antibody against *β*-actin (Abcam, ab8227) was purchased from Abcam (Cambridge, UK). 

The liver and kidney tissues (100 mg) were homogenized in RIPA lysis buffer containing protein inhibitors (Beyotime, Shanghai, China), and the supernatant was collected. Equivalent amounts of protein were separated by electrophoresis on 10% SDS-PAGE gels and transferred onto polyvinylidene fluoride (PVDF) membranes. The membranes were blocked with 5% fat-free milk and subsequently incubated with diluted primary antibodies. *β*-actin was used as the loading control. The membranes were then incubated with a goat anti-rabbit-HRP secondary antibody (1:5000, Servicebio, Wuhan, China). The protein-antibody complexes were detected using an enhanced chemiluminescence (ECL) system. The protein signals were analyzed and quantified using the ImageJ software.

### 2.6. Statistical Analysis 

SPSS 23.0 was used for statistical analysis. All the values are expressed as the mean ± standard deviation (SD). The Kolmogorov–Smirnov and Levene tests were used to check the data for assumptions of normality and homogeneity of variance, respectively, before performing statistical analyses. Three species were analyzed using one-way ANOVA, followed by an LSD test. The student’s *t*-test was used to evaluate the differences between the two altitudes. *p* < 0.05 was considered statistically significant. All charts were visualized using GraphPad Prism 8.0 (GraphPad Software, San Diego, CA, USA). 

## 3. Results

### 3.1. Serum Levels of 25(OH)D_3_, LCA, and Calcium and the Expression of Vitamin D_3_ Metabolism-Related Genes in Plateau Zokor, Plateau Pika, and SD Rats

As shown in Figure 1A, plateau zokors (3700 m) showed an undetectable serum level of 25(OH)D_3_, but the serum levels of 25(OH)D_3_ were significantly decreased in the plateau pikas (3700 m) compared with SD rats (1500 m) (Figure 1A). The serum levels of LCA in plateau zokors were significantly higher than those in the plateau pikas and SD rats (Figure 1B). Interestingly, the concentrations of calcium in the plateau zokors were within the normal range (1.10 ± 0.19 mmol/L), and there were no differences compared with the plateau pikas (1.10 ± 0.10 mmol/L) (Figure 1C). Compared with the SD rats (1.31 ± 0.02 mmol/L), the levels of calcium were significantly lower in plateau zokors and plateau pikas, but were still within the normal range (Figure 1C). 

*CYP2R1* is mainly expressed in the liver, whereas *CYP27B1* and *CYP24A1* are mainly expressed in the kidneys [2,13,54]. In the present study, the CYP2R1 mRNA and protein levels in the liver of plateau zokors (3700 m) were significantly higher than those in plateau pikas (3700 m) and SD rats (1500 m) (Figure 1D,G). *CYP27B1* mRNA levels in the kidney of plateau zokors were significantly higher than those of plateau pikas, and those of plateau pikas were significantly higher than those of SD rats (Figure 1E). CYP27B1 protein levels in the kidney of plateau zokors (3700 m) and SD rats (1500 m) were significantly higher than those of plateau pikas (3700 m) (Figure 1H), and there was no significant difference between the plateau zokors and SD rats (Figure 1H). *CYP24A1* mRNA levels in the kidneys of plateau pikas were significantly higher than those in plateau zokors and SD rats (Figure 1F). CYP24A1 protein levels in the kidneys of plateau pikas (3700 m) were significantly higher than those of plateau zokors (3700 m) and SD rats (1500 m) (Figure 1I), and there was no significant difference between the levels in plateau zokors and SD rats (Figure 1I).

### 3.2. Serum Levels of 25(OH)D_3_, LCA, and Calcium and the Expression of Vitamin D_3_ Metabolism-Related Genes in Plateau Zokors at Different Altitudes

Exposure to high-altitude hypoxia has various effects on the body, including the cardiovascular system, nervous system, and digestive system, which cause physiological and pathological changes. The plateau zokors at elevations of 3700 m and 2700 m had undetectable serum levels of 25(OH)D_3_ (Figure 2A). The serum levels of LCA in the plateau zokors significantly decreased with increasing altitude (Figure 2B). Interestingly, the calcium concentrations in plateau zokors were within the normal range (Figure 2C), and there was no significant difference between the levels in plateau zokors at elevations of 3700 m and 2700 m (Figure 2C). With increasing altitude, there were significant decreases in the levels of *CYP2R1*, *CYP27B1*, and *CYP24A1* mRNA in the liver and kidney, respectively, of plateau zokors (Figure 2D–F), and the levels of CYP2R1, CYP27B1, and CYP24A1 proteins were also decreased, but there were no significant differences (Figure 2G–I). 

### 3.3. Serum Levels of 25(OH)D_3_, LCA, and Calcium and the Expression of Vitamin D_3_ Metabolism-Related Genes in Plateau Pikas at Different Altitudes

With increasing altitude, the serum levels of 25(OH)D_3_ in plateau pikas significantly decreased (Figure 3A). However, serum levels of LCA and calcium did not differ significantly with increasing altitude (Figure 3B,C). With increasing altitude, the mRNA and protein levels of CYP2R1 in the liver and CYP27B1 in the kidney of plateau pikas significantly decreased (Figure 3D,E,G,H). In contrast, *CYP24A1* mRNA levels in the kidney of plateau pikas significantly increased (Figure 3F), whereas CYP24A1 protein levels were not significantly different (Figure 3I).

## 4. Discussion

The main source of vitamin D_3_ is cutaneous synthesis, for which sunlight is pivotal for photolysis of 7-dehydrocholesterol to vitamin D_3_. Vitamin D_3_ must undergo two subsequent hydroxylations, mainly by CYP2R1 and CYP27B1, to form the active form 1, 25(OH)_2_D_3_. Previous studies have shown that the subterranean rodents Damara mole-rats and naked mole-rats have a naturally impoverished vitamin D_3_ status [34,35]. The plateau zokor is a subterranean rodent that lives entirely underground in enclosed tunnels on the Qinghai-Tibet Plateau [55]. Similar to subterranean rodents, the plateau zokor lives in a hypoxic environment in the absence of sunlight. Our results showed that the plateau zokor was a vitamin D-deficient rodent with an undetectable serum level of 25(OH)D_3_. However, a previous study on Damara mole-rats and naked mole-rats showed that vitamin D_3_ supplementation or sun exposure resulted in an increase in serum level of 1, 25(OH)_2_D with a decline in 1-OHase activity and an increase in 24-OHase activity [34,35]. In the present research, we demonstrated that CYP2R1 mRNA and protein levels in the liver of plateau zokor were significantly higher than those of plateau pika and SD rats, and CYP27B1 mRNA and protein levels in the kidney of plateau zokor were significantly higher than those of plateau pika. *CYP24A1* encodes vitamin D 24-hydroxylase and is largely silent under vitamin D deficiency [56]. CYP24A1 mRNA and protein levels in the kidney of plateau pika were significantly higher than those of the plateau zokor and SD rats, although the protein levels were not significantly different between the plateau zokor and SD rats. Moreover, the CYP2R1 and CYP27B1 mRNA and protein levels in the liver and kidney decreased with increasing altitude, although the protein levels were not significantly different in the liver and kidney. Taken together, our results suggest that the plateau zokor is a vitamin D-deficient rodent, although the vitamin D_3_ metabolic enzymes CYP2R1 and CYP27B1 are highly expressed in the liver and kidney, respectively. These two enzymes may be related to the metabolism of other steroid hormones, which needs to be studied in the future.

It is well established that vitamin D_3_ is produced endogenously from 7-dehydrocholesterol following irradiation [21]. The Qinghai-Tibet Plateau receives strong ultraviolet radiation. In theory, the level of 25(OH)D_3_ in plateau pikas should be higher than that in SD rats. However, our results indicated that serum levels of 25(OH)D_3_ were significantly lower in plateau pikas than in SD rats. Moreover, 25(OH)D_3_ levels in plateau pikas significantly decreased with increasing altitude. A previous study reported severe vitamin D deficiency among nomads in Tibet [38]. However, biosynthesis of steroid hormones in SD rats is increased under acute hypoxia [36]. Another study showed that hypoxia increased the levels of 25(OH)D_3_ in sheep [37]. Therefore, our findings prompted us to investigate whether hypoxia regulates the expression of vitamin D_3_ metabolic genes. The critical steps in vitamin D_3_ metabolism are catalyzed by two enzymes CYP2R1 and CYP27B1 [54]. The results of our present study showed that the mRNA levels and protein abundances of CYP2R1 and CYP27B1 in the liver and kidney, respectively, of plateau pikas were significantly decreased in the high-altitude group compared with those in the low-altitude group. However, CYP24A1 protein abundance in the kidney of plateau pikas was not significantly different among the different altitude groups. Therefore, hypoxia can suppress the conversion of D_3_ into 25(OH)D_3_ and 1, 25(OH)_2_D_3_ by down-regulating the expression of *CYP2R1* and *CYP27B1* in plateau pika. Future research is needed to determine if hypoxia treatment suppresses the expression of *CYP2R1* and *CYP27B1* in rats or mice, which would enhance the understanding of the relationship between hypoxia and D_3_ metabolism.

Calcium homeostasis is crucial for the functioning of numerous health maintenance and biological processes [4]. In fact, the duodenum absorbs only 8–10% of dietary calcium [12]. Ishizawa et al. reported that LCA is a VDR ligand that acts preferentially in the ileum, and LCA did not increase plasma calcium concentrations or intestinal *TRPV6* expression in wild-type mice [57]. With regard to the absorption sites involved, the distal intestine might play a significant role in intestinal calcium absorption [9,10,11,12,13]. Studies have shown that calcium absorption is normal in vitamin D-deficient animals, including vitamin D-deficient rats and subterranean rodents such as Damara mole-rats and naked mole-rats [35,41,58]. Further, Nehring et al. reported that LCA restored serum calcium levels to the normal range by increasing VDR target gene expression and bone calcium mobilization in vitamin D-deficient rats [41]. Recently, intestinal calcium absorption was found to be increased by LCA in a vitamin D receptor-dependent manner [59]. The plateau zokor possesses morphological, behavioral, and physiological adaptations for digging, and they exhibit markedly powerful skeletal and muscle systems, lower pulse rates, and a stronger cardiovascular system compared with animals on the plain [46,60]. Despite the undetectable serum level of 25(OH)D_3_ in the plateau zokor, its calcium levels were within the normal range. Therefore, we speculated that there might have been a substance that replaced vitamin D_3_ in the plateau zokor. We also measured the LCA levels using UPLC-MS/MS, and the results showed that the LCA content of plateau zokors was significantly higher than that of plateau pikas and SD rats. These results suggest that LCA might be a substitute for vitamin D_3_ in the plateau zokor. As the most effective ligand of VDR, 1, 25(OH)_2_D_3_ promotes intestinal calcium absorption mainly through transcellular pathways mediated by TRPV6, calbindin-D_9k_, and plasma membrane Ca^2+^-ATPase 1b (PMCA1b), the expression of which is correlated with the calcium absorption efficiency [15,61,62]. However, Hashimoto et al. reported that LCA increased intestinal calcium absorption in a transcellular pathway-independent manner [59]. The physiological link connecting LCA and calcium metabolism in a naturally vitamin D-deficient rodent remains unclear. Therefore, the mechanisms underlying LCA-dependent calcium regulation in naturally vitamin D-deficient rodents requires further study.

## 5. Conclusions

In conclusion, the plateau zokor is a vitamin D-deficient subterranean rodent with an undetectable serum level of 25(OH)D_3_ as a result of the inhibition of the conversion of 7-dehydrocholesterol to vitamin D_3_, but its enzymes are highly expressed. Hypoxia significantly suppresses the conversion of D_3_ into 25(OH)D_3_ and 1, 25(OH)_2_D_3_ by down-regulating the expression of *CYP2R1* and *CYP27B1* in plateau pika. Further, LCA might be a substitution for D_3_ which could promote calcium absorption in the distal intestine of plateau zokors. However, the mechanisms underlying LCA-dependent calcium regulation in plateau zokors need to be identified.

## Figures and Tables

**Figure 1 animals-12-02371-f001:**
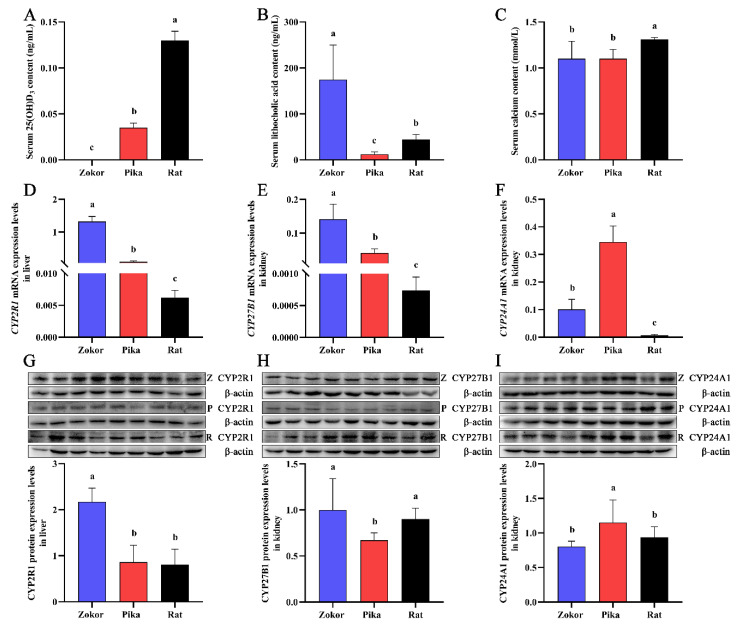
Serum levels of 25(OH)D_3_, lithocholic acid, and calcium and the expression levels of CYP2R1, CYP27B1, and CYP24A1 in the liver and kidney of plateau zokor, plateau pika, and Sprague-Dawley (SD) rats. (**A**–**C**) represent the serum levels of 25(OH)D_3_, lithocholic acid, and calcium, respectively. The mRNA (**D**) and protein (**G**) expression levels of CYP2R1 in the liver. The mRNA (**E**) and protein (**H**) expression levels of CYP27B1 in the kidney. The mRNA (**F**) and protein (**I**) expression levels of CYP24A1 in the kidney. Z, P, and R represent the plateau zokor at 3700 m elevation, plateau pika at 3700 m elevation, and SD rat at 1500 m elevation, respectively. Significant differences among species are represented by different lowercase letters (*p* < 0.05). No significant differences among species are represented by the same lowercase letters (*p* > 0.05). (original Western blot figures are in Appendix A).

**Figure 2 animals-12-02371-f002:**
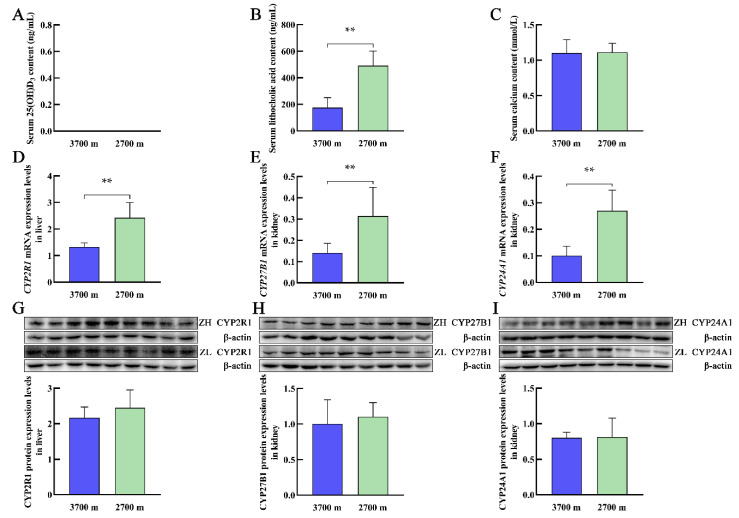
Serum levels of 25(OH)D_3_, lithocholic acid, and calcium and the expression levels of CYP2R1, CYP27B1, and CYP24A1 in the liver and kidney of plateau zokor at different altitudes. (**A**–**C**) represent the serum levels of 25(OH)D_3_, lithocholic acid, and calcium, respectively. The mRNA (**D**) and protein (**G**) expression levels of CYP2R1 in the liver. The mRNA (**E**) and protein (**H**) expression levels of CYP27B1 in the kidney. The mRNA (**F**) and protein (**I**) expression levels of CYP24A1 in the kidney. ZH and ZL represent the plateau zokor at elevations of 3700 m and 2700 m, respectively. ** *p* < 0.01. (original Western blot figures are in Appendix A).

**Figure 3 animals-12-02371-f003:**
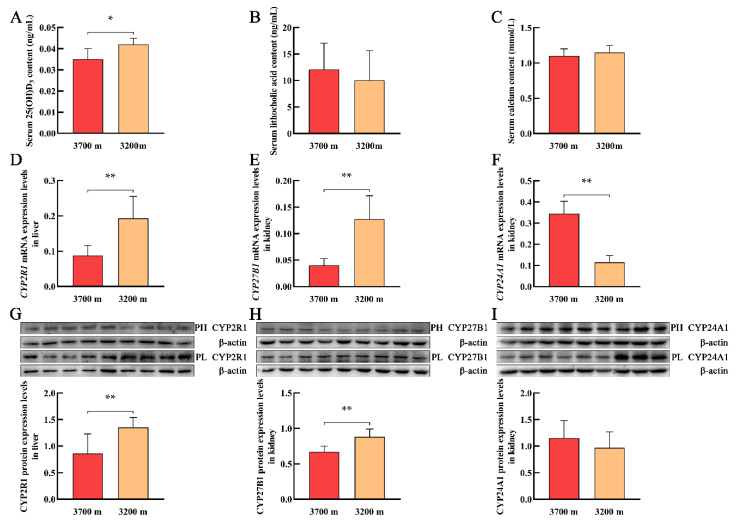
Serum levels of 25(OH)D_3_, lithocholic acid, and calcium and the expression levels of CYP2R1, CYP27B1, and CYP24A1 in the liver and kidney of plateau pika at different altitudes. (**A**–**C**) represent the serum levels of 25(OH)D_3_, lithocholic acid, and calcium, respectively. The mRNA (**D**) and protein (**G**) expression levels of CYP2R1 in the liver. The mRNA (**E**) and protein (**H**) expression levels of CYP27B1 in the kidney. The mRNA (**F**) and protein (**I**) expression levels of CYP24A1 in the kidney. PH and PL represent the plateau pika at elevations of 3700 m and 3200 m, respectively. * *p* < 0.05, ** *p* < 0.01. (original Western blot figures are in Appendix A).

**Table 1 animals-12-02371-t001:** Primers of vitamin D_3_ metabolism-related genes and *β*-actin for qPCR.

Species	Gene	Sense Primer (5′ to 3′)	Anti-Sense Primer (5′ to 3′)
Plateau Zokor	*CYP2R1*	AGTTGTTCAGTGAGAATGTGGAA	CCGAGGTAAGTGAGGCTTTC
	*CYP27B1*	CCAAGCCACTGTTCTATC	TAGTCACCTACACGGATG
	*CYP24A1*	GAGATTCGGACTCCTTCAGA	GGTGTTGAGCCTCTTGTG
Plateau Pika	*CYP2R1*	TTCCTCGGCAACATCTAC	CTACATCATAGCCATTCAGAAC
	*CYP27B1*	CAATGCTCTGTCTCAACTG	CGTGAAGTGGCATAGTGA
	*CYP24A1*	TCTCGCAATCCTCAAGTC	ACATATTCCTCAAGTCCTCTG
Sprague-Dawley rat	*CYP2R1*	GCATATCAACTGTGGTTCTCAAT	ATCCATCCTCTGCCATATCTG
	*CYP27B1*	ATGGTGAAGAATGGCAGAG	TGTCCAGAGTTCCAGCATA
	*CYP24A1*	TTCGCTCATCTCCCATTC	CATCTCCACAGGTTCATTG
	*β* *-actin*	TCACCAACTGGGACGATATG	GTTGGCCTTAGGGTTCAGAG

## Data Availability

Not applicable.

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
