# Peer review of "Vitamin D3 Metabolic Enzymes in Plateau Zokor (Myospalax baileyi) and Plateau Pika (Ochotona curzoniae): Expression and Response to Hypoxia"

_animals, 2022, doi:10.3390/ani12182371_

Round 1

Reviewer 1 Report

The study uses two unique animal species, the plateau zokor and plateau pika, as a model to investigate the effect of hypoxia on the metabolism of Vitamin D3. The authors detected the serum levels of 25(OH)D3, calcium, and LCA comparing zokor, pika, and rat. The 25(OH)D3 is undetectable in pika serum, and LCA concentration is significantly higher in the zokor serum. They also measured the expression level of D3 metabolism-related genes, and showed that the expression of CYP2R1 in the liver and CYP27B1 in the kidney of pika decreased with increasing altitude. The study reveals that hypoxia suppresses D3 in pika by suppressing the expression of D3 metabolism-related genes, and D3 is deficient in the zokor. The study provides informative models to understanding the relationship between D3 metabolism and hypoxia tolerance. The results are solid and it is interesting story.

minor comments:

1.  Figure 1 D-E: If hypoxia suppresses D3 metabolism by suppressing CYP2R1 and CYP27B1 genes (which means these two genes are responsible for the synthesis of D3), why would a naturally higher expression of these two genes in pika and zokor (Fig 1 D-E) not result in a higher level of D3 in these animals?

2. It would be interesting to directly test if hypoxia treatment (instead of different altitude) suppresses the expression of CYP2R1 and CYP27B1. This would enhance the understanding of the relationship between hypoxia and D3 metabolism. The experiment can be conducted in rat or mouse. This point could be discussed in the Discussion as a proposed future study.

3. D3 is closely related with the accessibility to sunlight, therefore, the absence of D3 in zokor may have resulted in both lack of sunlight exposure (due to subterranean habitat) and hypoxia (supported by this study). The combination of these two factors may even have a synergetic effect, which could have been discussed in the MS.

 4. Hope describes the time and location for animal samples collection at Materials and Methods  section, the authors collected samples and performed experiments at local regions or at lab? ( the rats from Lanzhou (1500 m) were kept and moved to Xining (2200 m) in advance one week for adaptation lab environment? And try to avoid variation, the different animal’s circadian rhythm and base line should be represented in Materials and Methods.

 5. The figs1-3 need to improve.

Reviewer 2 Report

animals-1873754

Chen and colleagues set out to investigate how animals that are rarely exposed to sunlight deal with the expected deficit in vitamin D levels and possible detrimental effects on calcium levels. Authors took advantage of two species of naturally occurring small mammals that differ in their biology in terms of exposure to sun in line with August Krogh principle “For such a large number of problems there will be some animal of choice or a few such animals on which it can be most conveniently studied.”. The author measured levels of key biomarkers related to vitamin D3 metabolism, noteworthy, enzymes involved in key pathways at both protein and mRNA levels in selected organs. The main finding was the confirmation of the authors’ prediction that the underground species would have very low levels of vitamin D and that LCA appears to substitute it. Overall, the manuscript is very well-written and appears very well conducted using a technically sound methodology and experimental design. Please see my few comments below:

1.    The authors highlight in both the introduction and discussion that vitamin D or steroids are upregulated in response to hypoxia in mammals. Still, it is not clear what are the possible implications of such response. The authors must address, even if briefly, the adaptive or functional significance of vitamin D (or LCA) in hypoxia tolerance. Do vitamin D or its associated metabolite play any role in hypoxia tolerance? This would strengthen the manuscript and also make it clear what was the rationale behind comparing animals living at different altitudes.

2.    The authors should briefly mention in the results section the implications of living at higher altitude to oxygen availability. The reader only finds its information at the end of the article in the methodology section and this information is vital to appreciate the results show in figures 2 and 3.

3.    Please confirm whether the differences between Pikas and Rats are actually significant in graphs 1C, and 1I. The same for Zokor x Pika in figure 1H.

Minor issues

4.    Page 2: “extremely active” is not very informative. Do the authors mean that biosynthesis of hormones is activated/induced in response to hypoxia in rats? Please be clear.

5.    Page 3: “, and SD rat were detected”. More than detected, they had their levels measured/quantified. I suggest using a better term than ‘detected’.

Reviewer 3 Report

This manuscript provides some novel insights into vitamin D status of the plateau zokor and the plateau pika and the effects of hypoxia on vitamin D metabolism in the zokors and pikas. The manuscript is well written. The authors provide adequate background information on the topic. The findings are clearly explained, and the authors draw appropriate conclusions based on their data.  

Section 2.1 and 2.2: The authors state that concentrations of plasma zokors were within the “normal” range. How did the authors determine what is considered “normal” serum calcium concentrations for the Zokor?  

Section 2.1; paragraph 2. The authors should specify that the protein expression of CYP2R1, CYP27B1, and CYP24A1 were measured in Zokor and Pika at high altitude.  

Section 4.3: “Arterial” is misspelled. Can the authors clarify if “arterial” is the word they intended? Shouldn’t it be “serum”? 

Figure 1: G, H, I: For an accurate comparison, it would have been ideal to have samples from Zokor, Pika, and Rat on the same membrane, rather than comparing between membranes. How was accurate normalization ensured, given that the groups were run on separate membranes? 

Round 2

Reviewer 1 Report

Dear Authors:

You have responded to the questions. However, there are still some detailed issues that need to be addressed: 

Figure 1-3:

There are too much space between panel letters and the plots. Panel letters (A, B, C…) should be moved closer to the Y axis. For western blot figures (Fig. G,H,I), labels for protein name can be moved to the right side of the gel. This way, the WB bands can be aligned with plots of other panels.

Figs need to be enlarged.

 Methods section:

Collecting samples was arranged at same duration (10:00 -12:00 or 14:00-16:00) for pika, zoker and lab rat to minimize circadian rhythm effects? If so, the description should be included in the Methods section.

Besides, the animals pre-adaptation was carried out with the same light/dark cycle in the lab (i.e. live-trapped, place in cage for 4-5 hours) ? this schedule should be included in the Methods section (not just in the response letter).
